# Barriers and Facilitators to Antimicrobial Stewardship in Antibiotic Prescribing and Dispensing by General Practitioners and Pharmacists in Malta: A Systematic Review

**DOI:** 10.3390/antibiotics14121181

**Published:** 2025-11-21

**Authors:** Brian Fenech, Daniel Gaffiero

**Affiliations:** Psychology, School of Science, College of Science & Engineering, University of Derby, Derby DE22 1GB, UK

**Keywords:** antimicrobial stewardship, antibiotic prescribing, socio-ecological model, COM-B model, Malta

## Abstract

Objective: Antimicrobial resistance (AMR) is a top ten threat to global public health, and Malta remains among the highest antibiotic-consuming countries in the European Union. This systematic review aimed to identify barriers and facilitators influencing antimicrobial stewardship in Malta, focusing on general practitioners (GPs) and pharmacists. Methods and Measures: Eligible studies included GPs and/or pharmacists practising in Malta and explored influences on prescribing and/or dispensing. Systematic searches were performed in June 2025 and September 2025 using the following databases MEDLINE, PsycINFO, PsycArticles PubMed, and Google Scholar. Data were extracted using a modified Cochrane template, and quality was assessed using Joanna Briggs Institute tools. Findings were synthesised using the socio-ecological model and mapped to the COM-B framework. Results: Seven studies met inclusion criteria, with a total sample size of 495 participants. Barriers included diagnostic uncertainty, knowledge gaps, misconceptions about AMR, patient expectations, commercial pressures, limited diagnostic and IT infrastructure, and defensive prescribing linked to indemnity insurance. Facilitators included stewardship values, stronger guideline adherence among younger GPs, trust-based GP–patient relationships, GP–pharmacist collaboration, and intervention effects from a social marketing programme. Mapping to COM-B showed barriers and facilitators interacting across capability, opportunity, and motivation. Conclusions: Prescribing in Malta is shaped by diagnostic uncertainty, entrenched habits, patient expectations, and structural gaps. Although the evidence base was limited and partly overlapping, consistent findings across mixed method designs highlighted that effective stewardship will require rapid diagnostics, e-prescribing, over-the-counter enforcement, and GP–pharmacist collaboration, supported by policy reforms aligning indemnity and sick-leave systems with AMR goals.

## 1. Introduction

Antimicrobial resistance (AMR) is one of the most pressing global health threats of the 21st century, undermining the treatment efficacy across antibiotics, antivirals, antifungals, and antiparasitic agents [1]. As resistance proliferates, routine infections become increasingly intractable whilst standard medical procedures, including surgery and chemotherapy, carry heightened risks, imposing longer hospital stays, escalating costs, and greater resource burdens on health systems [2]. In 2019, bacterial AMR was associated with an estimated 4.95 million deaths and directly attributable to approximately 1.27 million deaths worldwide [1,3], placing it among the leading causes of mortality globally. The World Health Organization [WHO] identifies misuse and overuse of antimicrobials across human health, animal husbandry, and environmental contexts as primary drivers of resistance [1].

Besides its health impact, AMR poses a profound economic challenge, with the World Bank projecting that unchecked resistance could reduce global gross domestic product by $1–$3.4 trillion annually by 2030, alongside substantial healthcare expenditures [4]. Recent European Commission briefings, drawing on new GRAM/Lancet forecasts, project that up to 39 million people could die from bacterial AMR between 2025 and 2050 [5]. These projections highlight AMR as an ongoing crisis that demands urgent behavioural and systemic interventions.

Within the European Union (EU) and European Economic Area (EEA), AMR causes over 35,000 deaths annually, comparable to the combined toll of influenza, tuberculosis, and HIV/AIDS [5]. In response, the Council of the EU issued a 2023 “One Health Recommendation” calling for at least a 20% reduction in total human antibiotic consumption by 2030 [6]. However, current use remains well above this target. In 2023, consumption averaged 20.0 defined daily doses (DDDs) per 1000 inhabitants per day against the 2030 goal of 15.9, with community-use alone standing at 18.3 DDD [7]. Moreover, the community broad-to-narrow spectrum ratio was 5.5, indicating broad-spectrum antibiotics were prescribed five times more frequently than narrow-spectrum agents. This prescribing pattern accelerates resistance by exposing far more non-target bacteria to selective antibiotic pressure [7].

Malta, with just over half a million residents [8], exemplifies how EU-level AMR targets must be translated into action within small, mixed public–private primary-care systems, where behavioural and cultural factors strongly shape prescribing patterns. Following modest declines between 2011 and 2019 and further reductions during 2020–2021, antibiotic use rebounded sharply in 2022 to the highest level in a decade [9]. By 2023, total consumption had risen to 22.9 DDD per 1000 inhabitants per day, with community-use alone standing at 20.9 DDD—placing Malta among the highest-consuming EU/EEA countries [9]. This prescribing profile is particularly concerning; the community broad-to-narrow ratio reached 20.8 in 2023, nearly four times the EU/EEA mean of 5.5, indicating an overreliance on broad-spectrum agents that amplifies resistance selection pressure [7]. Economic modelling projects that AMR will cause over 32 annual deaths in Malta and cost approximately €4 million per year, without stronger action [10].

Malta’s Strategy and Action Plan for the Prevention and Containment of AMR (2020–2028) adopts a “One Health” approach, coordinating surveillance, antimicrobial stewardship, diagnostics, infection prevention and control, and public education [11]. Implementation is overseen by the National AMR Committee and supported by initiatives including the Maltese Antibiotic Stewardship Programme in the community—a social marketing intervention aimed at general practitioners (GPs) [11,12]. Yet, persistent gaps between policy and practice demonstrate that technical guidelines alone cannot ensure prescribing decisions align with EU reduction targets.

These gaps reflect complex behavioural dynamics rather than simple knowledge deficits. A 2024 survey by Malta’s National AMR committee revealed that 55% of GPs reported reluctance to prescribe amoxicillin for community respiratory infections, believing local strains were resistant [13]. Yet surveillance data showed amoxicillin non-susceptibility in Streptococcus pneumoniae and Streptococcus pyogenes below 5%, directly contradicting this belief [13]. The briefing reiterated that amoxicillin remains widely available and is the recommended first-line treatment for uncomplicated bacterial respiratory infections. It urged clinicians to use diagnostic support, such as point-of-care strep testing, to distinguish viral from bacterial presentations. This disconnect illustrates how misinformed beliefs about local resistance drive prescribing toward broader-spectrum agents, even when narrower options remain effective, thereby reinforcing the resistance patterns that stewardship efforts seek to contain.

Socio-ecological pressures operating at multiple levels amplify this knowledge–practice gap. Eurobarometer 522 indicates that a significant minority of Europeans still expects antibiotics for colds and flu, with Malta among the highest-use countries—underscoring persistent public misconceptions and demand for “quick fixes” that clinicians must negotiate in time-limited encounters [14]. Maltese qualitative studies triangulating GPs, pharmacists, and parents consistently report diagnostic uncertainty, perceived patient expectations for antibiotics, and mixed acceptance of delayed prescriptions, particularly for acute respiratory tract infections [15,16,17]. These dynamics are intensified by structural features, including walk-in primary care without fixed GP registration, short consultation times, and frequent co-location of private GP services within community pharmacies [15,16,17]. Within this environment, capability constraints (e.g., limited access to rapid tests), opportunity constraints (e.g., restricted consultation time, scarce decision support), and motivational pressures (e.g., risk perceptions, defensive practice, prevailing social norms) converge to steer prescribing from guideline-concordant care [12,16,17].

Understanding these factors requires theoretical frameworks capturing multi-level influences and behavioural mechanisms. WHO/Europe recommends theory-informed approaches [18]. The COM-B conceptualises behaviour as interactions between capability (physical/psychological capacity), opportunity (physical and social opportunity factors), and motivation (automatic/reflective processes) [19], while the socio-ecological model influences across individual, interpersonal, organisational, community, and policy levels [20].

Whilst individual studies have explored Maltese prescribing and dispensing practices [12,15,16,17,21], no systematic synthesis has mapped the full range of barriers and facilitators shaping antimicrobial stewardship. Understanding how capability, opportunity, and motivational factors interact—particularly for GPs and pharmacists—remains essential for designing effective interventions.

This systematic review aims to (i) identify behavioural, systemic, and contextual factors influencing stewardship practices across primary care and community pharmacy settings; (ii) examine how capability, opportunity, and motivation interact to shape stewardship behaviours amongst GPs and pharmacists; (iii) generate evidence-based recommendations for adapting stewardship programmes to Malta’s mixed public–private system; and (iv) provide insights for other small European states facing comparable AMR challenges.

The review employs a dual theoretical approach: the socio-ecological model organises barriers and facilitators across system levels, whilst the COM-B captures underlying behavioural mechanisms. This complementary framework addresses the following research question: “What are the behavioural, systemic, and contextual barriers and facilitators influencing antimicrobial stewardship in antibiotic prescribing and dispensing among GPs and pharmacists in Malta?” Findings will inform interventions addressing both ecological context and behavioural mechanisms, supporting Malta’s 2030 antibiotic reduction targets whilst safeguarding quality of care.

## 2. Materials and Methods

### 2.1. Systematic Review Methodology

This review was conducted in accordance with the PRISMA (Preferred Reporting Items for Systematic Reviews and Meta-Analyses) guidelines to ensure methodological rigour. A comprehensive systematic literature search was undertaken to identify studies exploring barriers and facilitators to antimicrobial stewardship in antibiotic prescribing and dispensing in Malta. To enhance transparency and reduce risk of bias, the review protocol was prospectively registered in the PROSPERO database (Registration ID: CRD420251079452).

### 2.2. Inclusion and Exclusion Criteria

Studies were eligible for inclusion if they met the following criteria:i.Included GPs and/or pharmacists practicing in Malta.ii.Examined factors influencing antibiotic prescribing or dispensing, including attitudes, perceptions, knowledge, behaviours, barriers, or facilitators.iii.Conducted within the Maltese healthcare system (public or private), in primary care and/or hospital settings.iv.Employed a quantitative, qualitative, or mixed method design based on primary data.v.Published in a peer-reviewed journal in English or with an English translation available.vi.Published within the past ten years.

Studies were excluded if they:i.Did not include GPs or pharmacists as the primary population of interest (e.g., focused solely on nurses, patients, or specialists).ii.Were conducted outside of Malta.iii.Focused exclusively on clinical efficacy, microbiological resistance mechanisms, or intervention outcomes, without exploring behavioural or contextual influences on prescribing.iv.Addressed topics unrelated to prescribing practices (e.g., antimicrobial manufacturing, veterinary use, or supply chain logistics).v.Were reviews, editorials, opinion pieces, or commentaries without original empirical data.vi.Were not published in English and lacked a translated version.

### 2.3. Search Strategy

Systematic searches were conducted in June 2025 and repeated in September 2025 across MEDLINE, PsycINFO, PsycArticles PubMed, and Google Scholar. Search terms were informed by terminology used in a previous systematic review on barriers and facilitators to appropriate antibiotic-use interventions in low- and middle-income countries [22]. The strategy combined both Medical Subject Headings (MeSHs) and free-text terms to maximise sensitivity and ensure comprehensive retrieval.

The key search concepts included the following:i.Antimicrobial stewardship (e.g., “Antimicrobial Stewardship” [MeSH], “antimicrobial stewardship”, “antibiotic stewardship”),ii.GPs and pharmacists (e.g., “general practitioner*”, “family physician*”, “clinician*”, “pharmacist*”, “community pharmacist”),iii.Geographic location (e.g., “Malta”).

Search strategies were adapted for each database. Searches were limited to articles published in English (or with an English translation available) between June 2015 and June 2025 to ensure relevance to the current Maltese policy context. Additional relevant studies were identified by scanning the reference lists of included articles.

### 2.4. Search Results

The database searches yielded 258 articles. Of these, 217 were excluded at the title and abstract stage for not meeting the inclusion criteria, leaving 24 articles for full-text review. To ensure rigour, both authors (BF and DG) independently assessed article eligibility. Initial agreements were 87.5%, with three discrepancies resolved through discussion. Following consensus, seven studies were retained for inclusion.

Journal titles were cross-checked against the Directory of Open Access Journals (DOAJ) and the Committee on Publication Ethics (COPE) to ensure quality. No studies from journals listed on the Critical Appraisal Skills Programme (CASP) watchlist were included. The final review therefore comprised seven full-text papers (see Figure 1, PRISMA flow diagram and Table 1; study characteristics are below).

### 2.5. Data Extraction

Data extraction was conducted using a modified Cochrane Collaboration Template. Both authors independently extracted data from all included studies, then compared and reconciled results to ensure consistency. For quantitative studies, data items included study aims, methods, participant demographics, inclusion/exclusion criteria, withdrawals, subgroup information, sample sizes, outcomes related to barriers and facilitators, statistical findings, and key conclusions. For qualitative studies, data items included study aims, methodological approach (e.g., grounded theory, phenomenology), data collection and analysis methods, participant demographics, thematic findings on barriers and facilitators, and authors’ interpretations and conclusions.

### 2.6. Quality Assessment of Studies A–G

The methodological quality of the included studies was assessed using the Joanna Briggs Institute (JBI) critical appraisal tools, each applied according to study design. Analytical cross-sectional studies were evaluated with the JBI Checklist for Analytical Cross-Sectional Studies focusing on sample selection, measurement validity, and management of confounding variables. Quasi-experimental studies were assessed using the JBI Checklist for Quasi-Experimental Studies, which examines the clarity of cause–effect relationships, presence of a comparison group, and consistency of outcome measurement. Qualitative studies were appraised using the JBI Checklist for Qualitative Research, assessing methodological rigour, credibility, dependability, and relevance of findings.

Both authors (BF, DG) independently appraised all studies. Six discrepancies were identified and were resolved through discussion. Inter-rater agreement was high, with 83.33% observed agreement and Cohen’s Kappa = 0.71, indicating substantial agreement beyond chance. A detailed overview of the quality assessment is provided in Table 2 below.

Points were deducted for methodological shortcomings as follows: Study A lost points on Q5 and Q6, while GP age and sex were included as covariates, so no systematic strategy to control for confounding was reported (Q5), and contextual influences such as rurality and population age were acknowledged but not analysed (Q6). Study B lost points on Q1 and Q5–Q7: inclusion criteria were unclear (Q1), and confounders were neither identified (Q5) nor addressed (Q6), and although the validated APQ-Pharm questionnaire was used, internal consistency was not reported (Q7). Study E lost points on Q4 and Q6 for absence of a control group (Q4) and for failing to perform sensitivity analyses to assess attrition bias (Q6). Study F lost points on Q1 and Q7, as the philosophical perspective was not stated (Q1) and no reflexivity section was provided (Q7). Study G lost points on Q1 and Q6–Q7 due to the absence of a stated philosophical perspective (Q1), lack of researcher positioning (Q6), and omission of reflexivity (Q7).

## 3. Results

### 3.1. Demographic Information of Participants in the Included Studies

The review included seven studies reporting data from GPs, pharmacists, and parents in Malta. GP samples were reported in six studies (A, C, D, E, F, G). Sample sizes ranged from 8 to 112, with participants predominantly male, aged between 32 and 79 years, and reporting 0–49 years of practice experience. Although full details are presented in Table 1, a consistent pattern across studies was the predominance of private solo practice, with smaller proportions working in public health centres or in mixed-sector arrangements (A, C, D, F).

Several studies drew from overlapping cohorts. Study D (*n* = 30 GPs + 3 trainees; 4831 consultations) and Study E (*n* = 51 recruited; 33 provided pre-intervention data; 18 provided paired pre/post data) both recruited via the national GP surveillance project. Studies C and F interviewed 20 GPs between them in 2014, with identical demographic profiles, strongly suggesting overlap. Consequently, the total number of GP participants across studies should not be interpreted as unique individuals.

Pharmacist data were reported in two studies: 209 pharmacists participated in a national cross-sectional survey (Study B) and 24 pharmacists participated in focus groups (Study G). Pharmacists were mainly community-based; in Study B, 95/209 (45%) had more than 10 years of practice experience. Parents were included only in Study G (*n* = 18), with the majority being women in their 30 s and early 40 s.

### 3.2. Study Designs

Methodological approaches varied, meaning meta-analysis was not possible. Studies A, B, and D employed cross-sectional designs (Study A: Theory of Planned Behaviour (TPB) survey of GP prescribing intentions; Study B: national pharmacist survey; Study D: repeated surveillance of acute respiratory tract complaints (aRTCs); Study E used a quasi-experimental pre/post design to evaluate a multicomponent stewardship intervention embedded in the surveillance system; Studies C, F, and G employed qualitative methods (interviews or focus groups)) analysed through phenomenography, latent content analysis, or mixed inductive/deductive frameworks.

### 3.3. Aims of the Included Studies

The aims of each study were (A) to examine GP intentions to prescribe using the TPB; (B) to investigate the perception of Maltese pharmacists to prescribe a selected number of antibiotics; (C) to explore GPs’ views of delayed antibiotic prescribing for respiratory tract infections among Maltese GPs; (D) to identify factors influencing GPs’ oral antibiotic prescribing for aRTCs; (E) to evaluate the impact of a social marketing stewardship intervention on GP prescribing; (F) to investigate barriers and facilitators to prudent prescribing for aRTCs; and (G) to capture perspectives of GPs, pharmacists, and parents on the prescribing–use–dispensing dynamic.

### 3.4. Barriers and Facilitators to Prudent Prescribing and Dispensing

Synthesis of studies A–G identified barriers and facilitators to antimicrobial stewardship across the five socio-ecological levels: individual, interpersonal, organisational, community, and policy. A full summary of barriers and facilitators is presented in Table 3.

#### 3.4.1. Individual Level

Barriers: Prescribing was shaped by expectations to prescribe, willingness to prescribe without clear diagnostic evidence, and habitual behaviours such as repeat or third-party prescriptions (Study A). Older age, longer experience, and female sex were associated with higher prescribing (Study D). Diagnostic uncertainty was a consistent driver, reinforced by beliefs that viral infections often progressed to bacterial infections (Study F). Additional barriers included inconsistent dosing practices, preference for broad-spectrum agents, post-intervention class shifts (e.g., increased tetracycline use; Study E), pharmacists’ decreased confidence in their knowledge for prescribing certain antibiotic classes (e.g., metronidazole, quinolones, Study B), and patient behaviours such as non-compliance, self-medication, and unsafe disposal (Study G).

Facilitators: The majority of pharmacists endorsed protocol-based prescribing (Study B). Younger GPs were more guideline-compliant than older GPs (Study G). Intervention evidence showed reduced prescribing and greater uptake of delayed prescriptions, though interrupted time series analyses indicated that improvements were not sustained in the long term (Study E). GPs also expressed stewardship-oriented attitudes, describing antibiotics as “precious” and citing strategies such as delayed prescribing, observation periods, tailoring regimens, and risk-based prescribing (Study F).

#### 3.4.2. Interpersonal Level

Barriers: Social pressures were strong. Subjective norms were the most powerful predictor of intention to prescribe (Study A). Fear of patient loss (“doctor shopping”; Study C), pressure to satisfy regular clients (Study F), and direct patient requests (Studies F, G) contributed to unnecessary prescribing. Patient misconceptions (e.g., fever requires antibiotics, antibiotics as anti-inflammatories) complicated consultations, and some GPs admitted conceding to patient pressure (Study F, G). Professional isolation, particularly among solo private GPs, and occasional pharmacist pressure were also noted (Study F).

Facilitators: Established GP–patient relationships and clear communication supported acceptance of delayed prescriptions (Study C). Trust in pharmacists, coupled with specific advice, encouraged alternatives such as symptomatic management (Study G). GPs viewed patient education as a professional duty, noting gradual improvements in public awareness (Study F). Collaborative GP–pharmacist relationships, where advice was shared, were seen as beneficial (Studies F, G).

#### 3.4.3. Organisational Level

Barriers: Private pharmacy-based GPs prescribed more than those in public health centres, albeit this did not reach statistical significance (*p* = 0.063; Study D), which is a difference plausibly linked to commercial pressures and patient expectations. Lack of diagnostic testing, limited access to POCTs, time delays, and costs restricted evidence-based prescribing (Study F, G). Weak IT infrastructure, poor communication between care levels, disorganised services, and limited consultation time further encouraged defensive prescribing (Studies F).

Facilitators: The social marketing intervention produced modest but significant shifts in prescribing behaviour in clustered analyses, though not sustained in time series models (Study E). GPs expressed willingness to adopt POCTs if reliable (Study F) and called for access to community data, updated guidelines, and electronic prescribing systems as enablers (Study G).

#### 3.4.4. Community Level

Barriers: Cultural norms supported precautionary prescribing, with uncertainty avoidance cited as an influence (Study F). Persistent misconceptions of antibiotics as a cure-all and inconsistent practices in delayed prescribing (C, F, G) undermined stewardship. Pharmaceutical marketing, including promotion of certain classes or higher doses, was also influential (Study F).

Facilitators: Stakeholders acknowledged overuse and recognised rising public awareness (Study G). Patients showed increasing willingness to accept delayed prescriptions and symptomatic care, especially when supported by clear advice and education from GPs (Studies C, F, G). Younger GPs’ stronger adherence to guidelines also reflected a cultural shift within the profession, with pharmacists and colleagues observing greater receptiveness to evidence-based practice and strategies such as delayed prescribing among newer generations (Study G).

#### 3.4.5. Policy Level

Barriers: Introduction of professional indemnity insurance in 2014 was linked to more defensive prescribing (Study F). Inconsistent enforcement allowed some over-the-counter (OTC) antibiotic sales to persist, despite reported decline (Study G).

Facilitators: Stricter enforcement of OTC bans and proposals to reform sick leave policies—enabling parents to stay home with sick children without relying on antibiotics—were identified as supportive measures (Study G).

### 3.5. Bridging to COM-B

Mapping the findings onto the COM-B framework provides a behavioural diagnosis that complements the socio-ecological analysis. This approach highlights how prescribing and dispensing practices in Malta are shaped by interacting domains of capability, opportunity, and motivation.

Psychological capability was limited by gaps in knowledge, diagnostic uncertainty, and misperceptions among both clinicians and patients (Studies B, F). Examples included beliefs that viral infections routinely progress to bacterial ones and patient misconceptions such as the idea that fever requires antibiotics (Study F). Physical capability was similarly constrained by restricted access to point-of-care diagnostics, insufficient training, and limited consultation time that reduced opportunities for patient education (Study F, G). By contrast, facilitators of capability included younger GPs’ greater adherence to guidelines, stewardship values among some clinicians, and strong communication practices that supported patient understanding (Studies E, F, G).

Opportunity was also critical. Social opportunity was shaped by patient demand, peer norms, professional isolation, and wider cultural drivers of antibiotic use, all of which pushed prescribers toward precautionary or unnecessary prescribing (Studies C, F, G). Facilitators at this level included trust in GP–patient relationships, positive pharmacist–client interactions, collaborative working between GPs and pharmacists, and a generational shift toward more guideline-concordant practice among younger GPs (Studies D, G). Physical opportunity was constrained by structural barriers such as commercial pressures in private practice, weak enforcement of over-the-counter sales bans, inadequate diagnostic and IT infrastructure, and the influence of pharmaceutical marketing (Study F). Conversely, facilitators included greater access to reliable point-of-care tests, use of national antibiotic guidelines and surveillance data, electronic prescribing systems, and consistent regulatory enforcement (Studies, E, F, G).

Motivation emerged as a particularly influential domain. Reflective motivation was undermined by entrenched prescribing habits, defensive practices linked to indemnity insurance, and intentions shaped by previous behaviour and commercial considerations (Studies A, D, F). Yet reflective motivation was also strengthened by stewardship-oriented attitudes, recognition that most acute respiratory tract infections are viral, and willingness among many practitioners to adopt delayed prescribing strategies (Studies C, F, G). Automatic motivation was driven by fear of patient deterioration, habitual reliance on broad-spectrum antibiotics, and automatic concession to patient requests (Studies F, G). Nonetheless, facilitators were evident, including the ability of experienced GPs to resist patient pressure (D) and evidence that stewardship interventions show some short-term promise in disrupting habitual prescribing by reducing overall use and increasing uptake of delayed prescriptions (Study E).

Taken together, this behavioural diagnosis indicates that stewardship interventions in Malta cannot rely solely on education or awareness-raising. Instead, they must act synergistically across COM-B domains, strengthening capability through targeted training and communication, creating opportunity through diagnostics, IT systems, and robust regulation, and sustaining motivation by embedding stewardship values and reshaping professional and social norms. A detailed summary of the mapped barriers and facilitators across all six COM-B domains is presented in Table 4.

## 4. Discussion

The present systematic review addressed the question: “What are the barriers and facilitators to prudent antibiotic prescribing and dispensing in Malta?”. Seven studies were synthesised, spanning the perspectives of general practitioners and pharmacists [12,15,16,17,21,23,24]. Using the socio-ecological model [25,26] barriers and facilitators were identified across multiple levels of influence, and subsequent mapping onto the COM-B framework [27] provided a behavioural diagnosis of how capability, opportunity, and motivation interact to shape prescribing. This dual framework approach offered both contextual breadth and behavioural depth, strengthening the implications for intervention design. However, the interpretation of findings below is primarily framed through the COM-B model. This is because COM-B provides a behavioural diagnosis that explains how barriers and facilitators operate through capability, opportunity, and motivation, whereas the socio-ecological model situates where these influences occur.

### 4.1. National Measures for Antibitoic Stewardship in Malta

Before interpreting the findings of the present systematic review, it is important to situate them within Malta’s ongoing national efforts to strengthen antimicrobial stewardship. The National Antibiotic Committee (NAC) plays a central role in implementing the Strategy and Action Plan for the Prevention and Containment of AMR in Malta (2020–2028). Its work includes developing and disseminating national antibiotic prescribing guidelines and conducting educational campaigns targeting the public and healthcare professionals. Moreover, the University of Malta has introduced a blended intensive programme, “Antimicrobial Stewardship: Managing Antibiotic Resistance”, which includes sessions on behaviour change, interventions, patient and civil-society engagement, and leadership, policy and political commitments to combat AMR. Taken together, these developments indicate significant progress in aligning Malta’s stewardship efforts with the EU One Health Action Plan. However, our review indicates that gaps remain between policy and routine prescribing and dispensing practices.

### 4.2. Interpretation of Findings

The synthesis highlights that capability constraints remain central to inappropriate prescribing. Across surveys and qualitative studies, diagnostic uncertainty, gaps in knowledge of specific antibiotic classes, and limited salience of antimicrobial resistance emerged as consistent barriers [15,17,21,23,24]. These patterns mirror international research, identifying diagnostic uncertainty as a leading driver of precautionary prescribing in primary care [28,29]. At the same time, stewardship values and stronger guideline adherence among younger GPs [16,24] point to attitudinal and generational shifts that could be harnessed through targeted training and professional development.

Opportunity factors, both social and physical, were equally influential. Social opportunity was strongly shaped by patient expectations, cultural attitudes positioning antibiotics as a “cure-all”, and professional isolation, reinforcing findings from low-to-middle income settings where cultural drivers and patient demand similarly influence prescribing [30]. However, relational facilitators such as trust-based GP–patient relationships, collaborative GP–pharmacist partnerships, and increasing public awareness of AMR [16,23,24] demonstrate the potential of communication and partnership to counterbalance demand. Physical opportunity was constrained by commercial pressures in private practice, limited diagnostics and IT infrastructure, and weak enforcement of over-the-counter restrictions [16,23,24]. These structural issues echo broader stewardship implementation challenges reported internationally [31], including in European contexts such as England and Sweden [32]. At the same time, the demand for updated guidelines, electronic prescribing, and accessible surveillance data [16,24], alongside intervention evidence showing modest behaviour change from social marketing [12], highlight practical opportunities to strengthen system-level supports.

Motivational processes also played a key role. Reflective motivation was shaped by entrenched habits, defensive prescribing linked to indemnity insurance, and beliefs about disease progression [21,23,24]. Automatic motivation reflected fear of deterioration, habitual broad-spectrum prescribing, and concession to patient requests [16,23,24]. These findings resonate with behavioural science evidence that prescribing is often influenced by defensive reasoning and affective responses rather than purely rational decisions [33,34]. Encouragingly, stewardship attitudes, recognition that most respiratory tract infections are viral, and the capacity of some experienced GPs to resist pressure [24], as well as evidence of disrupted habits following interventions [12], point to motivational levers that could be strengthened.

### 4.3. Practical Implications for Malta

To provide practical direction for policymakers, the COM-B mapping was translated into a set of prioritised implications for Malta. At the immediate level, introducing and subsidising rapid POCTs would directly address diagnostic uncertainty and reduce precautionary prescribing [23,24]. Expanding GP and pharmacist training in stewardship, communication skills, and DAP should be targeted especially at older GPs—who were found to prescribe more habitually—and at pharmacists who reported confidence gaps in certain antibiotic classes [15,17,21,24]. National promotion of structured DAP, using standardised pads and patient information leaflets, could further support clinician confidence and patient acceptance [12,16,17].

In the medium term, stewardship would benefit from the implementation of a national e-prescribing system, enabling real-time audit and feedback as well as stronger GP–pharmacist collaboration [16,24]. Enforcement of regulations prohibiting over-the-counter antibiotic sales should be reinforced, accompanied by pharmacy-based disposal bins and public education on safe disposal [16,24]. Regular updating of prescribing guidelines, including local community-level data, is also necessary to increase GP trust and uptake [16,24].

Finally, system-level reforms should be considered. Revising professional indemnity and sick-leave policies may help reduce defensive prescribing and patient pressure for rapid return to work or school [16,24]. Insurer guidance should explicitly affirm guideline-concordant non-prescribing as defensible standard of care and align cover with stewardship aims. Public education and social marketing campaigns should continue but will need stronger top-down support and integration with IT and diagnostic improvements to achieve sustained change [12,16]. Expanding collaborative models between GPs and pharmacists, including protocol-based pharmacist prescribing for minor ailments under national frameworks, could extend stewardship capacity while maintaining quality [15,16]. A full prioritised implications table, structured by tier and mapped to COM-B domains, is provided in Table 5 below.

### 4.4. Limitations and Future Research Directions

This review has certain limitations that should be considered when interpreting the findings. The evidence base was relatively small, and several studies drew on overlapping GP cohorts—for example, the same 20 GPs interviewed in 2014 informed both Saliba–Gustafsson et al. [17] and Saliba–Gustafsson et al. [24], while the intervention cohort reported by Machowska et al. [12] was recruited from the surveillance study of Saliba–Gustafsson et al. [23]. This overlap reduces the effective participant pool. However, the reuse of cohorts also enabled complementary analyses across different designs (qualitative and quantitative), revealing important and nuanced insights into Maltese GPs’ prescribing practices and attitudes that may not have been captured through a single study design alone.

Another limitation concerns the included studies employing varied quantitative and qualitative designs, which restricted causal inference and precluded meta-analysis. Finally, most included studies were conducted before or during the early phases of Malta’s 2020–2028 national stewardship strategy. Peer-reviewed evaluations since the strategy’s launch remain sparse, so the most recent prescribing and dispensing trends, and the effects of national initiatives, are not yet well-characterised in the Maltese literature. Malta needs more longitudinal monitoring of GP prescribing and pharmacy dispensing, alongside real-world evaluation studies in primary care and community pharmacy settings, so that how programmes are delivered, how accessible they are, and what difference they make can be assessed within the current 2020–2028 horizon and used to shape the post-2028 plan. Despite these limitations, the review retains considerable value. The consistency of findings across different study designs, settings, and participant groups strengthens the conclusions presented in this review. Moreover, the dual use of the socio-ecological model and COM-B provided explanatory depth that compensates for methodological heterogeneity, yielding actionable insights for stewardship policy and practice in Malta.

Future research should prioritise longitudinal and mixed method studies to track how prescribing and dispensing behaviours evolve as Malta’s national stewardship strategy matures. Greater attention to pharmacists’ roles and patient perspectives would provide a more comprehensive picture of influences across the health system. Evaluations of interventions—such as point-of-care testing, audit and feedback, or public campaigns—are also needed to establish their effectiveness and sustainability in Malta’s mixed public–private primary-care context. Comparative research with other small European states could further clarify which behavioural and structural levers are transferable and which require tailoring to Malta’s health system.

## 5. Conclusions

This systematic review highlights that the barriers and facilitators to prudent antibiotic prescribing and dispensing in Malta are multi-layered, spanning individual, social, organisational, community, and policy levels. Using the socio-ecological model alongside the COM-B framework, we identified the levels at which barriers and facilitators occur, and the potential mechanisms through which they influence behaviour. Effective stewardship interventions must therefore act across domains by building capability through education, creating opportunity through systemic reform, and sustaining motivation through cultural and professional change. Integrated behavioural and structural approaches offer the strongest potential to reduce unnecessary prescribing and should directly inform Malta’s next One Health strategy beyond 2028.

## Figures and Tables

**Figure 1 antibiotics-14-01181-f001:**
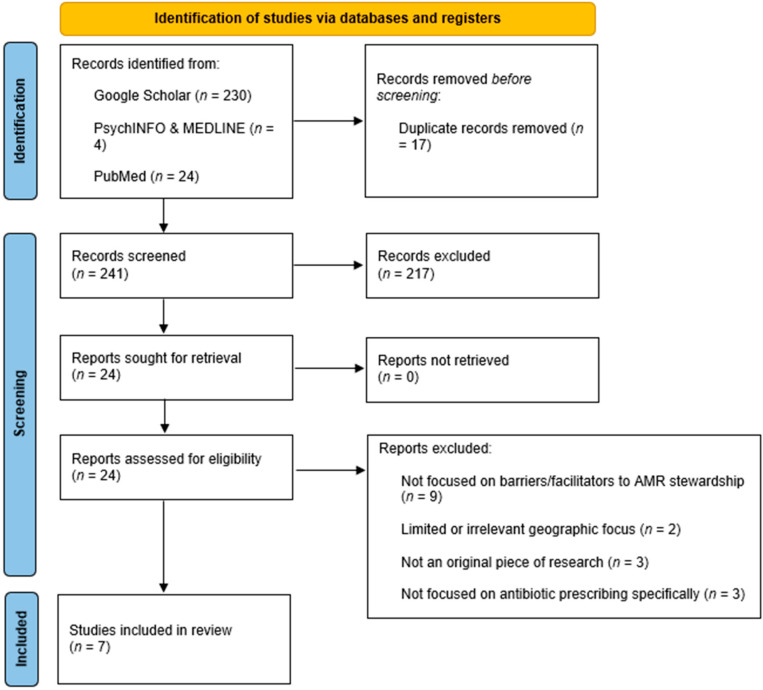
PRISMA flow diagram illustrating the study selection process for barriers and facilitators to antimicrobial stewardship in antibiotic prescribing in Malta.

**Table 1 antibiotics-14-01181-t001:** Characteristics of the included studies.

Study	Design	Participants	Demographics	Materials/Measures	Statistical Tests	Qualitative Analysis	Outcome(s) Measured
**A** **Tsiantou et al.** [21]	Cross-sectional questionnaire design	General practitioners (*n* = 112) from Malta located in morthern and southern harbour areas, southeastern and western areas)	Females (*n* = 42, 38%) Mean age, years (46 ± 1.1) Years of experience: <5 (8%), 5–10 (9%), >10 (83%) Practice location: rural (0%), semi-urban (32%), urban (68%) Practice type: public (11%), private (66%), combination (23%) Single/group practice: single (62%), group practice/health centre/hospital (38%)	Demographic Questionnaire Theory of Planned Behaviour Questionnaire (designed by the authors) and translated into Maltese (and available in English)	Pearson’s chi-squared independence test Kruskal–Wallis hypothesis test (parametric/non-parametric) Cronbach’s Alpha Multiple linear regression	N/A	Intention to prescribe Intention to prescribe without well-documented evidence Generalised intention to prescribe GP prescribing behaviour in the recent past (four questions)
**B** **Attard Pizzuto et al.** [15]	Cross-sectional questionnaire design	Pharmacists practising in Malta (*n* = 209)	Years of experience: >10 (*n* = 95) Employed in community pharmacies (*n* = 87), locum pharmacists (*n* = 34), own private pharmacy (*n* = 28) Time spent checking and dispensing antibiotic prescriptions: 21–40 h (38%) Dispensing > 1 antibiotic prescription daily (72%)	Demographic questionnaire Antibiotic Prescribing by Pharmacists Questionnaire (APQ_Pharm_) developed by the authors	Friedman Test	N/A	Pharmacists’ antibiotic knowledge and practice Pharmacists’ agreement and competency to prescribe antibiotics
**C** **Saliba-Gustafsson et al.** [17]	Qualitative Phenomenographic study	Registered general practitioners (*n* = 20)	Male (*n* = 14, 70%) Age: 30–49 (*n* = 2), 40–49 (*n* = 4), 50–59 (*n* = 11), 60–69 (*n* = 2), 70–79 (*n* = 1) Years of experience in general practice: 0–9 (*n* = 2), 10–19 (*n* = 3), 20–29 (*n* = 10), 30–39 (*n* = 3), 40–49 (*n* = 2) Healthcare sector of practice: public (*n* = 4), private (*n* = 14), both (*n* = 2)	Demographic Questionnaire Semi-structured interview guide	N/A	Data were analysed using the phenomenographic approach. Steps taken: (1) familiarisation, (2) compilation and condensation, (3) comparison, grouping and preliminary description, (4) formulation and labelling of different categories of description, (5) final categorisation of descriptions and outcome space	Interview topics included views on antibiotic resistance, antibiotic use for respiratory tract infections, and barriers and facilitators to antibiotic prescriptions
**D** **Saliba-Gustafsson et al.** [23]	Repeated cross-sectional surveillance design	General practitioners (*n* = 30) and trainee general practitioners (*n* = 3) registered data of 4831 patients of all ages suffering from any acute respiratory tract infection	General practitioners Male (*n* = 24, 73%) Mean age: 49 ± 12 Mean years of GP practice: 23 ± 11 Healthcare sector: public (*n* = 11, 33%), private (*n* = 20, 61%), both sectors (*n* = 2, 6%) Patients Female (*n* = 2395, 53.1%) Median age 29 (IQR = 12–48) Education: completed up to secondary school (*n* = 3050, 68%) Smoker (*n* = 735, 16.5%)	Questionnaire to record demographics, training/experience and service delivery organisation Forms during surveillance weeks for first consultation with patients of all ages suffering from any acute respiratory tract infection including patient and clinical factors, clinical assessment, diagnosis and prescribed medicines	Complete-case analysis Population-averaged models using generalised estimating equations Multivariate Wald-type tests	N/A	Factors associated with antibiotic prescribing
**E** **Machowska et al.** [12]	Repeated cross-sectional design with intervention elements	General practitioners (*n* = 51), pre-intervention completers (*n* = 33), post-intervention (*n* = 18)	Pre-intervention Age, median (52, IQR = 24–57) Sex: Male (*n* = 24) Years of GP practice (23, IQR: 16–29) Type of practice: Group (*n* = 15), Solo (*n* = 18) Employment sector: public (*n* = 11), private (*n* = 20), both (*n* = 2) Employment type: full-time (*n* = 22), part-time (*n* = 11) Post-intervention Age, median (46, IQR = 41–53) Sex: Male (*n* = 13) Years of GP practice (20, IQR: 15–24) Type of practice: group (*n* = 10), Solo (*n* = 8) Employment sector: public (*n* = 8), private (*n* = 9), both (*n* = 1) Employment type: full-time (*n* = 14), part-time (*n* = 4)	The intervention had five key elements: Patient booklets (six pages) available in English and Maltese. Patient waiting room posters (*n* = 4) Soft and hard copies of the updated national antibiotic guidelines Standardised DAP pads including patient information on respiratory tract Infections and appropriate antibiotic use in English and Maltese Educational sessions (*n* = 4) of a duration of 2 h held online and in-person	Mann–Whitney U Pearson’s chi-quare Fisher’s exact test Population-averaged models using generalised estimating equations Interrupted time series analysis	N/A	Change in antibiotic prescription for acute respiratory tract conditions Change in antibiotic prescription for immediate use, for delayed antibiotic prescription, by diagnosis, and antibiotic class
**F** **Saliba-Gustafsson et al.** [24]	Qualitative: Latent content analysis	General practitioners (*n* = 20)	Male (*n* = 14, 70%) Age: 30–39 (*n* = 2), 40–49 (*n* = 4), 50–59 (*n* = 11), 60–69 (*n* = 2), 70–79 (*n* = 1) Experience in general practice (years): 0–9 (*n* = 2), 10–19 (*n* = 3), 20–29 (*n* = 10), 30–39 (*n* = 3), 40–49 (*n* = 2) Health sector of practice: public (*n* = 4), private (*n* = 14), both (*n* = 2)	Demographic Questionnaire Semi-structured interview guide	N/A	Manifest and latent content analysis using an inductive approach Transcripts were read independently by two authors; meaning units were identified and condensed into meaning units to identify codes; codes were grouped to develop themes/sub-themes; findings were then visualised in a socio-ecological model	How GPs’ behaviour is influenced by processes and interactions at the individual, interpersonal, organisational, community, and public policy levels
**G** **Saleh et al.** [16]	Qualitative: Inductive and deductive content analysis	General practitioners (*n* = 8), pharmacists (*n* = 24), parents (*n* = 18)	GPs Male (*n* = 6, 75%) Age range: FG1: 50–70 (*n* = 6), FG2: 41–59 (*n =* 2) Pharmacists Female (*n* = 19, 79.17%) Age range: FG3: 25–44 (*n* = 7), FG4: 25–56 (*n* = 11), FG5: 25–65 (*n* = 6) Parents Female (*n* = 13; 72.22%) Age range: FG6: 36–43 (*n* = 5), FG7: 30–46 (*n* = 5), FG8: 29–40 (*n* = 8)	Demographic questionnaire Semi-structured focus group discussion guides	N/A	Data were analysed using deductive and inductive content analysis Data were triangulated to understand stakeholders’ perspectives Transcripts were read several times; meaning units were selected and condensed to generate codes; codes were grouped into sub-categories and coalesced into categories	Antibiotic use and antibiotic resistance in Malta from the perspectives of GPs, pharmacists, and parents Influence of interpersonal relationships among patients, GPs, and pharmacist on antibiotic use Solutions for action—tackling antimicrobial resistance in Malta

*Note*: GP = general practitioner; FG = focus group; N/A = not applicable.

**Table 2 antibiotics-14-01181-t002:** Quality assessment of included studies.

Study	Checklist	# of Items	Original BF Score	Original DG Score	Number of Discrepancies	Agreed Score	Quality Score (%)	Quality Rating
**A** **Tsiantou et al.** [21]	JBI Checklist for Analytical Cross-Sectional Studies	8	6	6	0	6	75	High
**B** **Attard Pizzuto et al.** [15]	JBI Checklist for Analytical Cross-Sectional Studies	8	4	5	1	4	50	Medium
**C** **Saliba–Gustafsson et al.** [17]	JBI Checklist for Qualitative Research	10	10	10	0	10	100	High
**D** **Saliba–Gustafsson et al.** [23]	JBI Checklist for Analytical Cross-Sectional Studies	8	7	7	2	8	100	High
**E** **Machowska et al.** [12]	JBI Checklist for Quasi-Experimental Studies	9	8	7	1	7	77.78	High
**F** **Saliba–Gustafsson et al.** [24]	JBI Checklist for Qualitative Research	10	9	7	2	8	80	High
**G** **Saleh et al.** [16]	JBI Checklist for Qualitative Research	10	7	7	0	7	70	Medium

*Note*: JBI = Joanna Briggs Institute; # = number.

**Table 3 antibiotics-14-01181-t003:** Full list of barriers and facilitators to prudent antibiotic prescribing in Malta.

Level	Barriers	Facilitators	Supporting Studies
**Individual**	Prescribing habits and intentions (A) Diagnostic uncertainty (C, F) Belief viral infections progress to bacterial (F) Persistent broad-spectrum use (E) GP demographics: older age, longer experience, female sex (D) Pharmacist decreased confidence in prescribing certain antibiotic classes (metronidazole, quinolones) (B) Patient behaviours: non-compliance, self-medication, unsafe disposal (G)	Endorsement of protocol-based prescribing among the majority of pharmacists (B) Stewardship values (F, G) Recognition that most URTIs are viral (F) Use of delayed prescriptions (E, F) Younger GPs more guideline-compliant (G) Intervention effects: decreased prescriptions for immediate use, increased delayed antibiotic prescribing (E) Availability of antibiotic disposal bins in pharmacies (G)	A, B, C, D, E, F, G
**Interpersonal**	Social pressure/subjective norms (A) Fear of losing patients (“doctor shopping”) (C) Pressure to satisfy regular clients (F) Direct patient requests for antibiotics (F, G) Patient misconceptions (fever requires antibiotics; antibiotics as anti-inflammatories, antibiotics as “holy grail”) (F, G) Resistant parents (G) Quick GP access increasing prescribing (G) Professional isolation (solo GPs) (F) Occasional pharmacist pressure (F)	Trust and continuity in GP–patient relationships (C, G) Positive pharmacist–client interactions with specific advice (G) Clear communication strategies to support delayed prescribing (C, G) GP-led patient education; gradual improvement in awareness (F) Experienced GPs resisting patient pressure (D) Collaborative GP–pharmacist relationships (F, G)	A, C, D, F, G
**Organisational**	Higher prescribing in private pharmacy-based clinics (D) Lack of diagnostics/POCTs; time and cost barriers (F) Weak IT infrastructure; poor communication between sectors; disorganisation of services (F) Limited awareness and use of guidelines (F) Short consultation times restricting patient education (F)	Social marketing intervention: modest but positive prescribing shifts (E) Willingness to adopt POCTs if reliable/efficient (F) Calls for local data and updated guidelines (F, G) Desire for electronic prescribing and health records (G)	D, E, F, G
**Community**	Cultural uncertainty avoidance driving precautionary prescribing (F) Antibiotics perceived as “cure-all” (G) Variability in delayed prescribing practices (G) Pharmaceutical marketing influence (promotion of classes/doses) (F)	Recognition of antibiotic overuse and declining effectiveness (G) Growing public awareness and acceptance of delayed prescriptions, especially with advice (C, G) Younger GPs showing stronger adherence to guidelines (G)	C, D, F, G
**Policy**	Defensive prescribing linked to indemnity insurance (F) Inconsistent enforcement of OTC sales ban (G)	Stricter enforcement of OTC bans (G) Proposed reforms to sick leave policies to reduce parental pressure for antibiotics (G)	F, G

*Note*: Levels of influence: individual = prescriber or patient knowledge, attitudes, habits, beliefs, demographics; interpersonal = GP-patient, GP-pharmacist, pharmacist-client interactions; organisational = healthcare setting, clinic resources, diagnostics, IT systems; community = cultural, societal, and marketing influences; policy = regulatory or structural drivers at national level. Abbreviations: URTI = upper respiratory tract infection; POCT = point-of-care testing; OTC = over-the-counter sales.

**Table 4 antibiotics-14-01181-t004:** Mapping of barriers and facilitators to the COM-B model.

Domain	Barriers	Facilitators
**Psychological Capability**	Knowledge gaps in antibiotic classes (B); misconceptions such as the belief that viral infections progress to bacterial infections (F); patient misconceptions about antibiotics (F).	Younger GPs’ stronger adherence to guidelines (D, G); recognition that most URTIs are viral (F); stewardship values among clinicians (F); improved public awareness (G).
**Physical Capability**	Diagnostic uncertainty in the absence of POCT (F); lack of training in or use of diagnostics (F); limited consultation time for patient education (F).	Effective communication strategies (F, G); GP-led education initiatives (F); willingness to adopt POCT when available (F).
**Social Opportunity**	Patient pressure and subjective norms (A, F, G); fear of losing patients (C, F); resistant parental attitudes (G); professional isolation (F); pharmacist pressure (F, G); cultural drivers of antibiotic use (F); uncertainty avoidance (F, G).	Trust within GP–patient relationships (F); positive pharmacist–client interactions (G); collaborative relationship between GP–pharmacist (F, G); patient acceptance of delayed prescribing when supported by education (F, G); cultural shift toward guideline adherence among younger GPs (D, G).
**Physical Opportunity**	Commercial pressures in private practice (F); weak enforcement of OTC restrictions (G); limited diagnostics and IT infrastructure (F); poor coordination between healthcare sectors (F, G); influence of pharmaceutical marketing (F); time and cost barriers (F).	Updated prescribing guidelines (D, F); access to reliable POCT (D, E, F); availability of local surveillance data (D); enforcement of OTC bans (G); pharmacy disposal bins for antibiotics (G); multicomponent social marketing interventions (E).
**Reflective Motivation**	Prescribing habits and intentions shaped by prior behaviour (A, D); defensive prescribing linked to indemnity insurance concerns (F); commercial considerations in private practice (D, F).	Willingness to adopt delayed prescribing (C, F, G); professional commitment to patient education (F).
**Automatic Motivation**	Fear of deterioration or complications (F); habitual reliance on broad-spectrum antibiotics (F); automatic concession to patient requests (F, G).	Experienced GPs resisting patient pressure (D); evidence of short-term behaviour change following stewardship interventions (E).

*Note*: Abbreviations: AMR = antimicrobial resistance; URTI = upper respiratory tract infection; ARTI = acute respiratory tract infection; POCT = point-of-care testing; OTC = over-the-counter sales. Definitions: Reflective motivation refers to conscious decision-making processes, whereas automatic motivation refers to habitual, emotional, or reactive drivers of behaviour.

**Table 5 antibiotics-14-01181-t005:** Prioritised implications for antimicrobial stewardship in Malta, mapped to COM-B domains.

Tier	Action	COM-B Domain(s)
**Immediate priorities**	Introduce and subsidise rapid point-of-care tests to reduce diagnostic uncertainty and precautionary prescribing.	Psychological Capability; Physical Opportunity
	Expand GP and pharmacist training in communication skills, delayed antibiotic prescribing, and stewardship principles, with targeted focus on older GPs and pharmacists with knowledge gaps.	Psychological Capability; Reflective Motivation
	Promote structured delayed DAP through standardised pads and patient information leaflets to support professional confidence and patient acceptance.	Reflective Motivation; Social Opportunity
**Medium-term actions**	Implement a national e-prescribing system linked to electronic health records to enable audit/feedback and strengthen GP–pharmacist collaboration.	Physical Opportunity; Social Opportunity
	Reinforce enforcement of regulations prohibiting over-the-counter antibiotic sales, combined with pharmacy-based disposal bins and education on safe disposal.	Physical Opportunity; Reflective Motivation
	Update and disseminate national prescribing guidelines regularly, ensuring inclusion of community-level resistance and prescribing data.	Psychological Capability; Reflective Motivation
**System-level reforms**	Review professional indemnity and sick-leave policies to reduce defensive prescribing and patient pressure for rapid return to work or school.	Automatic Motivation; Reflective Motivation
	Sustain public education and social marketing campaigns, integrated with information technology (IT) and diagnostic improvements, and supported by national leadership.	Reflective Motivation; Social Opportunity
	Foster GP–pharmacist collaborative models, including protocol-based pharmacist prescribing for minor ailments under clear national frameworks.	Social Opportunity; Psychological Capability

*Note*: Abbreviations: COM-B, Capability, Opportunity, Motivation–Behaviour framework; DAP, delayed antibiotic prescribing; GP, general practitioner; IT, information technology. “Tiers” indicate the relative feasibility and timeframe of implementation. Immediate priorities represent high-feasibility actions that can be implemented quickly. Medium-term actions require infrastructure or system investments and are expected to take longer to achieve. System-level reforms involve structural or cultural changes that demand sustained commitment and policy-level support.

## Data Availability

Data extraction sheets are available upon reasonable request.

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
