# Peer review of "Barriers and Facilitators to Antimicrobial Stewardship in Antibiotic Prescribing and Dispensing by General Practitioners and Pharmacists in Malta: A Systematic Review"

_antibiotics, 2025, doi:10.3390/antibiotics14121181_

Round 1

Reviewer 1 Report

Comments and Suggestions for Authors

It is very interesting well written review
Authors should write Conclusion in a sigle format.
Authors should contain in a separate paragraph any measures have been taken in Malta in order to minimize antibiotic stewardship.

Author Response

Reviewer Comment: Authors should write Conclusion in a single format.

Author response: We thank the reviewer for this observation. The Conclusion section has now been revised to align with the journal’s preferred format. Specifically, it now succinctly summarises the key findings, theoretical contribution, and practical implications of our review. Please see lines 558-577.

Reviewer Comment: Authors should contain in a separate paragraph any measures have been taken in Malta in order to minimize antibiotic stewardship.

Author response:  In response, we have now added a new subsection titled “4.0 National Measures for Antibiotic Stewardship in Malta” at the beginning of the Discussion section. This paragraph summarises the key national initiatives implemented to promote prudent antibiotic use, including the establishment of the National Antibiotic Committee (NAC), which has led educational campaigns targeting both the public and healthcare professionals, and developed national antibiotic prescribing guidelines. We have also noted the University of Malta’s blended intensive programme, “Antimicrobial Stewardship: Managing Antibiotic Resistance”, which incorporates training on behaviour change, intervention design, civic engagement, and policy leadership to strengthen capacity in antimicrobial stewardship. Please see lines 450-463.

Reviewer 2 Report

Comments and Suggestions for Authors Originality and relevance: This is the first systematic review synthesizing evidence on antimicrobial stewardship in Malta using dual theoretical frameworks (socio-ecological model and COM-B). Given Malta's high antibiotic consumption rates within the EU/EEA, the work addresses a relevant gap. Methodology: The search strategy is comprehensive, quality appraisal rigorous, and synthesis appropriately handles heterogeneity across study designs. Limitations are transparently acknowledged. Conclusions: Well-supported by evidence and directly address the research question. Table 5 effectively translates findings into actionable policy recommendations. References and figures: Appropriate and clearly presented.

This is a well-structured and methodologically sound systematic review. Although only seven studies were ultimately included, the authors have synthesised the available evidence comprehensively and effectively. The use of the socio-ecological model and the COM-B framework adds substantial conceptual clarity, making the results easy to follow and insightful for both research and policy audiences. The manuscript is overall well written and coherent. My only suggestion is that the Introduction section is excessively long; it could be condensed by at least half to improve focus and readability.

Author Response

Reviewer comment: My only suggestion is that the Introduction section is excessively long; it could be condensed by at least half to improve focus and readability.

Author response: We thank the reviewer for this comment. We agree, the introduction is rather long. As such, we have significantly condensed this section to improve focus and readability. Please see lines 34-453.

Reviewer 3 Report

Comments and Suggestions for Authors

This is a very well written manuscript, of a systematic review exploring factors affecting prescribing practices in Malta, as presented through psychosocial and behavioral models.

The methodology is sound and presented in detail. The results are well described appropriately follow the methods and conform to the study aims and outcomes. The narrative parts of the manuscript are complete and discuss findings in relation to the literature, also providing local implications, future directions and limitations.

In general, I only have minor comments.

Methods

Please explain why included studies should be published within the last 10 years.

Figure 1

Please correct the box in the flowchart to remove spell-check.

 Results

Its not clear to me why the 7 studies are labelled A-G, when they could just be cited.

Author Response

Reviewer comment: Please explain why included studies should be published within the last 10 years.

Author response: The decision to include studies published within the last 10 years was made to ensure that the review accurately reflects the current state of antibiotic prescribing and dispensing practices in Malta. Over the past decade, Malta has undergone significant changes in antimicrobial stewardship policy, public awareness, and healthcare delivery. Including older studies risked capturing data collected before these systemic changes, which could limit the relevance of the findings.

Reviewer comment: Figure 1 - Please correct the box in the flowchart to remove spell-check.

Author response:  We thank the reviewer for noticing this. We have amended this accordingly.

Reviewer comment: Results - It’s not clear to me why the 7 studies are labelled A-G, when they could just be cited.

Author response: Thank you for this comment. We appreciate the suggestion. We chose to label the seven included studies as A–G to improve readability and reduce repetition throughout the Results section and tables. Because the same studies are referenced multiple times across different domains and COM-B constructs, labelling them A–G avoids repeated long citations in the narrative and prevents the tables from becoming text-heavy. We have also added studies' citations in Table 2 after their corresponding A-G labelling.